

# Improved method for linear carbon monoxide simulation and source attribution in atmospheric chemistry models illustrated using GEOS-Chem v9

Jenny A. Fisher[1,2], Lee T. Murray[3], Dylan B.A. Jones[4,5], and Nicholas M. Deutscher[1]

[1]Centre for Atmospheric Chemistry, School of Chemistry, University of Wollongong, NSW, Australia
[2]School of Earth and Environmental Sciences, University of Wollongong, NSW, Australia
[3]Department of Earth and Environmental Sciences, University of Rochester, Rochester, NY, USA
[4]Department of Physics, University of Toronto, Toronto, ON, Canada
[5]Joint Institute for Regional Earth System Science and Engineering, University of California, Los Angeles, CA, USA

*Correspondence to:* Jenny A. Fisher, jennyf@uow.edu.au

**Abstract.** Carbon monoxide (CO) simulation in atmospheric chemistry models is frequently used for source-receptor analysis, emission inversion, interpretation of observations, and chemical forecasting due to its computational efficiency and ability to quantitatively link simulated CO burdens to sources. While several methods exist for modelling CO source attribution, most are inappropriate for regions where the CO budget is dominated by secondary production rather than direct emissions. Here,
we introduce a major update to the linear CO-only capability in the GEOS-Chem chemical transport model that for the first time allows source-region tagging of secondary CO produced from oxidation of non-methane volatile organic compounds. Our updates also remove fundamental inconsistencies between the CO-only simulation and the standard full chemistry simulation by using consistent CO production rates in both. We find that relative to the standard chemistry simulation, CO in the original CO-only simulation was overestimated by more than 100 ppb in the model surface layer and underestimated in outflow regions. The improved CO-only simulation largely resolves these discrepancies by improving both the magnitude and
location of secondary production. Despite large differences between the original and improved simulations, however, model evaluation with the global dataset used to benchmark GEOS-Chem shows negligible change to the model's ability to match the observations. This suggests that the current GEOS-Chem benchmark is not well suited to evaluate model changes in regions influenced by biogenic emissions and chemistry, and expanding the dataset to include observations from biogenic source regions
(including those from recent aircraft campaigns) should be a priority for the GEOS-Chem community. Using Australasia as a case study, we show that the new ability to geographically tag secondary CO production provides significant added value for interpreting observations and model results in regions where primary CO emissions are low. Secondary production dominates the CO budget across much of the world, especially in the southern hemisphere, and we recommend future model-observation and multi-model comparisons implement this capability to provide a more complete understanding of CO sources and their
variability.





# 1 Introduction

Carbon monoxide (CO) has long been considered an excellent tracer for air mass origin. It has large direct sources from incomplete combustion and a lifetime of weeks-months that is sufficiently long for transport on intercontinental scales, but sufficiently short for enhancements to be distinct from the atmospheric background. CO emitted directly from combustion

sources is straightforward to model in chemical transport models (CTMs) and chemistry-climate models (CCMs) as its only chemical loss is through reaction with OH. As long as the OH distribution and chemical sources are known (i.e., extracted from a full chemical simulation) or can be parameterized, then the concentration of CO can be recalculated with an ordinary differential equation that is linear in the CO concentration and not coupled to other time-varying species. This linear CO simulation is highly computationally efficient, avoiding the need to solve the stiff chemical equations that dominate resource

use in CTMs and CCMs (Duncan et al., 2007). Emitted CO can also easily be "tagged" by source region or type and followed as it is transported to receptor regions downwind, providing a quantitative metric for source-receptor influence. This capability to determine air mass origin, along with the efficiency of CO-only simulation, means the linear CO capability has frequently been applied in a variety of CTMs and CCMs to interpret in situ observations of CO and co-varying species (e.g., Staudt et al., 2001; Liang et al., 2004; Fisher et al., 2010; Pfister et al., 2011), analyze satellite data (e.g., Park et al. 2009; Kumar et al. 2013),

improve emission estimates (e.g., Kopacz et al. 2010; Jiang et al. 2011), disentangle multi-model ensembles (e.g., Monks et al. 2015; Zeng et al. 2015) and forecast expected chemical conditions for field campaigns.

In regions where primary emissions dominate, the standard method of tagging CO based on primary emissions generally provides an accurate picture of the sources contributing to regional atmospheric composition. However, in much of the world (and especially in the southern hemisphere), direct CO emissions are small, localized, and/or episodic, and secondary produc-

tion dominates the CO budget (Zeng et al., 2015). In these regions, source attribution becomes more challenging. Models have addressed the problem of attribution for secondary CO in different ways. The simplest method is to treat secondary production as the difference between total CO and CO from primary emissions (e.g., Zeng et al., 2015). This method provides limited additional information as it cannot distinguish between source regions or differentiate between long-lived (e.g., methane) versus short-lived (e.g., isoprene, monoterpene) precursor sources. A second method has been to track the carbon from emission of a

precursor species (e.g., isoprene), distinguished if desired by region, through all intermediate products until it eventually ends up as CO (Pfister et al., 2008). This is by far the most accurate way to identify source influence for secondary CO, but it adds computational expense and is technically challenging to implement, especially as chemical mechanisms become increasingly complex. As a result, CO attribution using this method tends to lag advances in standard model mechanisms, and such attribution studies are not standard in any major atmospheric chemistry model. In principle, adjoint and similar sensitivity methods

can also be used for source attribution (Zhang et al., 2009); however, we are unaware of any study that has explicitly used such methods to constrain source influence for secondary CO, in part because co-location of primary and secondary sources can make it difficult to reliably distinguish between the two (Jiang et al., 2011).

The GEOS-Chem linear CO-only simulation has historically employed a different method (Duncan et al., 2007). Emissions of a subset of known CO precursors (isoprene, acetone, methanol, and monoterpenes) are assumed to instantaneously produce





CO. Precursor emissions are thus scaled by an assumed CO yield, and the CO produced is treated simply as an additional CO emission. Production from methane is treated similarly, but with CO scaled to the methane mixing ratio rather than methane emissions. While this method provides a decent approximation in the northern mid-latitudes where primary emissions dominate the CO budget (Duncan et al., 2007), it has several important flaws, particularly when applied to remote regions. First, it

assumes that all CO production occurs in the model surface layer, when in reality much of the production will be in the free troposphere. This is problematic because the surface layer is often decoupled from the free troposphere, resulting in potentially different transport pathways depending on source altitude (Leung et al., 2007; Chen et al., 2009). This vertical offset will impact longer-lived precursors that are oxidised during transport rather than immediately upon emission. Methanol, for example, has a lifetime of several days and represents up to 20% of the CO source in remote regions (Wells et al., 2014), but in the original

GEOS-Chem treatment is assumed to only produce CO in the continental boundary layer.

In addition, the current method decouples the source attribution capability of GEOS-Chem from the standard tropospheric chemistry simulation. Frequent updates and improvements to the chemical mechanism, driven largely by new understanding of biogenic chemistry (e.g., Mao et al. 2013; Fisher et al. 2016; Travis et al. 2016) impact CO production in ways that are non-linearly related to the abundance of precursors. These updates are not straightforward to translate to the CO simulation, which

relies on fixed yields from precursor oxidation. Updating the CO yields to match the standard chemistry simulation would require first applying the carbon tracing method described above, a time-consuming process that has not to our knowledge been performed for any recent version of GEOS-Chem.

The differences between the linear CO-only simulation and the standard full chemistry simulation are especially problematic when the two simulations are used in concert to interpret observations. Relative to the full chemistry simulation, secondary

CO in the CO-only simulation is produced with the wrong yield and at the wrong altitude. This disconnect between the two simulations complicates use of the CO-only simulation for attribution of co-varying species with potential biogenic origin (e.g., ozone, secondary organic aerosols). The vertical offset of secondary CO production may also be problematic for interpreting satellite CO observations. In the current CO-only simulation, CO that should be produced in the free troposphere is instead being emitted into the boundary layer, where many CO retrievals have limited sensitivity (Kopacz et al., 2010). This creates

an apparent artificial bias in the CO-only simulation (in addition to any actual bias in the full chemistry simulation) that could bias results of satellite-constrained inversions (Jiang et al., 2011).

Here we introduce a major update to the GEOS-Chem linear CO-only simulation that provides a more reliable source attribution capability. This method retains the computational expediency of the CO-only simulation while introducing full alignment with the standard full chemistry simulation. In the following sections, we first describe the new method (Sect. 2)

and compare the resulting global distribution of CO to the original CO-only and full chemistry simulations (Sect. 3). We then evaluate the original and improved CO-only simulations and the full chemistry simulation against a suite of global CO observations (Sect. 4). Finally, we use the Australasian region as a case study, comparing the original and improved simulations against observations from the Total Carbon Column Observing Network (TCCON) to explore the benefits of the improved source attribution capability for regions with limited impact from primary CO emissions (Sect. 5).



## 2  Model description

### 2.1  GEOS-Chem version, inputs, and experimental set-up

We use as base version the GEOS-Chem CTM version 9-01-03, but the method described here is easily translatable to more recent versions with minor updates to the code, and we have recently implemented it in a provisional version of v11-01

(v10-01 did not include the CO-only capability). GEOS-Chem is driven by assimilated meteorology from the Global Modeling and Assimilation Office (GMAO) Goddard Earth Observing System (GEOS-5). The native horizontal resolution of GEOS-5 is 0.5°x0.667°, with 3-hourly (2-D surface fields) or 6-hourly (3-D fields) temporal resolution.

In subsequent sections, we compare CO from three GEOS-Chem simulations: a (1) "standard full chemistry" run that simulates the full suite of chemical species in the GEOS-Chem mechanism (including CO) and their coupled production and

loss; (2) an "original CO-only" run that uses the standard CO-only capability included in GEOS-Chem v9-01-03 (described below); and (3) an "improved CO-only run that incorporates the modifications described in this work. For all simulations, we use a horizontal resolution of 2°x2.5° with model timesteps of 15 minutes for transport and 60 minutes for chemistry. Biomass burning CO emissions are from GFEDv3 and anthropogenic emissions are from EDGARv3.2 (fossil fuels) and Yevich and Logan (2003) (biofuels), with regional overwrites as detailed in Fisher et al. (2010). We simulate a three-year period (2009-

2011) to understand the influence of interannual variability on our results.

### 2.2  CO-only simulation in GEOS-Chem

Linear CO-only simulation (also known as "Tagged CO") has been a capability included in the GEOS-Chem CTM since version 2-08 (http://acmg.seas.harvard.edu/geos/geos_versions.html). The code for the CO-only simulation (as well as other speciality simulations) is embedded in the standard GEOS-Chem code, with the choice of simulation selected at runtime.

Hence, version numbers for the CO-only simulation are the same as for the full GEOS-Chem release. In what follows, the "original" CO-only simulation refers to v9-01-03, our base version.

The CO simulation operates as follows. In a given model grid box, the rate of change in the CO concentration ([CO]) due to emissions and the chemistry is given by:

$$\frac{d[CO]}{dt} = E + P(CO) - k[OH][CO] \tag{1}$$

where $E$ represents surface emissions, $P$(CO) represents chemical production of CO from methane and non-methane volatile organic compounds (NMVOCs), $k$ represents the rate constant for oxidation of CO by OH, and [OH] is the OH concentration. In Eqn. 1 we neglected transport to and from neighboring grid boxes since our focus here is on the influence of the chemistry, but the effects of advection are accounted for in the model. Surface emissions come from external inventories, as described in Sect. 2.1. In the standard GEOS-Chem simulation, $P$(CO) and $k$[OH][CO] in Eqn. 1 would be coupled to the concentrations

of other chemical species (methane, NMVOCs, OH, each represented by similar differential equations) using a chemistry solver. In contrast, the CO-only simulation is run in a chemically offline mode, in which the simulation is decoupled from the chemistry solver.





To calculate the chemical loss term ($k$[OH][CO]), monthly mean [OH] fields are saved from a prior run of the standard full chemistry simulation. Previously, the CO-only simulation used OH from earlier versions of the model (e.g., v5-v7 in Kopacz et al., 2010; Jiang et al., 2011; Fisher et al., 2010) to mitigate a known OH high bias in more recent versions of GEOS-Chem. Here we use OH from v9-01-03 in both our original and improved CO-only simulations to maintain consistency with the full

chemistry model and to ensure all changes are due to the new representation of chemical production rather than differences in OH. Both original and improved versions use the pressure- and temperature-dependent OH+CO rate constant ($k$) from the JPL data evaluation.

The chemical production term ($P$(CO)) requires special treatment as it depends on the time-varying concentrations of methane and NMVOC precursors. This term is treated differently in the original and improved versions of the CO-only simu-

lation, as described in detail in Sects. 2.2.1 and 2.2.2.

We made a few other small updates to the CO-only simulation that are included in both the original and improved versions compared below. We implemented optional non-local mixing in the planetary boundary layer (Lin and McElroy, 2010) and the centralized chemistry timestep (http://wiki.seas.harvard.edu/geos-chem/index.php/Centralized_chemistry_time_step) – both of which are defaults in the v9-01-03 standard full chemistry simulation but were missing from the CO-only simulation. We also

added diurnal scaling of the monthly mean [OH] fields based on the cosine of the solar zenith angle. This is the same method used in all other offline simulations in GEOS-Chem, and is an objective improvement to use of a single monthly mean value in each grid box. All of these updates are now implemented in the current public version of the GEOS-Chem linear CO-only simulation, v11-01.

### 2.2.1    Original CO-only simulation

In the original CO-only model, CO is produced chemically from 5 precursors: methane, acetone, methanol, isoprene, and monoterpenes. In other words, $P$(CO) = $P$(CO)$_{methane}$ + $P$(CO)$_{acetone}$ + $P$(CO)$_{methanol}$ + $P$(CO)$_{isoprene}$ + $P$(CO)$_{monoterpenes}$. For methane, the model assumes a 100% molar CO yield from oxidation by OH, with methane concentrations defined as averages of surface observations from NOAA carbon cycle surface flasks over four latitudinal bands (30-90°N/S, 0-30°N/S). These assumptions are applied throughout the troposphere.

For NMVOC precursors, CO production occurs in the model surface layer only (i.e., it is treated as an additional emission), with assumed molar yields of 67% from acetone, 100% from methanol, 30% from isoprene, and 20% from monoterpenes (Duncan et al., 2007). Isoprene and monoterpene emissions are from the Model of Emissions of Gases and Aerosols from Nature (MEGAN), as in the standard full chemistry simulation. Acetone emissions come from Jacob et al. (2002), and the methanol flux is scaled to the isoprene flux assuming a methanol:isoprene molar ratio of ~1:4. Acetone and methanol are

therefore decoupled from the full chemistry simulation and do not take into account recent updates to their budgets (e.g., Fischer et al. 2012; Wells et al. 2014). CO production from NMVOCs emitted during combustion is accounted for by increasing primary CO emissions from anthropogenic and biomass burning sources by 19% and 11%, respectively (Duncan et al., 2007). As for the biogenic NMVOC source, this production is applied directly in the surface layer.





### 2.2.2 Improved CO-only simulation

Our approach in the improved model is to couple $P$(CO) in the CO-only model to the standard chemistry model, using a 2-step approach similar to that applied in the Model for Ozone and Related chemical Tracers (MOZART-4) by Park et al. (2013). We first complete a run of the full chemistry simulation, from which we save monthly mean CO chemical production rates (in molecules cm$^{-3}$ s$^{-1}$). These rates are then used as inputs to calculate production in the CO-only simulation. While this method requires an extra step, the full chemistry run need only be completed once for every new GEOS-Chem model version, and this can easily be completed as part of the standard benchmarking procedure for new public releases of the model (http://acmg.seas.harvard.edu/geos/geos_benchmark.html). For many applications, use of a single model year (e.g., that used in the 1-year benchmarking process) will be sufficient; however, the code is designed in such a way that an interested user could easily re-run the standard chemistry simulation to save the CO production fields for a specific year of interest.

The total $P$(CO) is split into contributions from methane ($P$(CO)$_{CH4}$) and from NMVOCs ($P$(CO)$_{NMVOC}$) by also saving the methane loss rates ($L$(CH$_4$)) from the full chemistry simulation. We maintain the assumption from the original CO-only simulation of a 100% CO yield from methane oxidation, such that $P$(CO)$_{CH4}$ = $L$(CH$_4$). The NMVOC contribution is then the difference between the total CO production and the methane contribution: $P$(CO)$_{NMVOC}$ = $P$(CO) - $P$(CO)$_{CH4}$. The assumption of a 100% CO yield will somewhat overestimate the contribution from methane relative to the contribution from NMVOCs. This mainly affects the tropical lower troposphere, where we occasionally find that methane loss exceeds total CO production, likely due to rapid vertical transport of intermediate products. Scavenging of soluble intermediate products (for example, methyl hydroperoxide) also reduces the CO yield from methane, although this is expected to be a small effect (e.g., <0.1% in our 1-day tests for methyl hydroperoxide). In the few cases where $P$(CO)$_{CH4}$ overestimates total $P$(CO), we cap $P$(CO)$_{CH4}$ at the total CO production rate. This assumption does not affect the total CO production, retaining consistency with the standard full chemistry simulation.

The saved CO production fields $P$(CO)$_{CH4}$ and $P$(CO)$_{NMVOC}$ are then used as input to the linear CO-only model, similar to the approach used for the GEOS-Chem tagged odd oxygen simulation (Li et al., 2002). The archived fields are scaled diurnally using the cosine of the solar zenith angle, as done for OH concentrations in GEOS-Chem and for source profiles in other models (e.g., the model intercomparison of Zeng et al., 2015). Our updated treatment is applied in a flexible manner that allows a user to choose the original CO production method (scaling surface emissions) if so desired.

### 2.2.3 Source attribution capability

In both the original and improved simulations, Eqn. 1 is linear in [CO]. As a result of this linearity, CO from different sources and regions can be treated independently (referred to as tagged CO tracers; Bey et al., 2001). For primary emitted CO, Eqn. 1 becomes

$$\frac{d[CO_{i,j}]}{dt} = E_{i,j} - k[OH][CO_{i,j}] \tag{2}$$

where $i$ represents the emission type (e.g. fossil fuel, biomass burning), $j$ the geographical region in which that source is emitted, and [CO$_{i,j}$] is a separate CO tracer for each emission type and geographical region.





For secondary chemically produced CO, the treatment differs between the two model versions. In the original model, secondary CO is distinguished by the precursor NMVOC but not by region, such that Eqn. 1 becomes

$$\frac{d[CO_{k,global}]}{dt} = P(CO)_{k,global} - k[OH][CO_{k,global}] \tag{3}$$

where $k$ represents the precursor (i.e. acetone, methanol, isoprene, or monoterpenes). In the improved model, we can no longer distinguish between different NMVOCs. Instead, we now tag secondary CO from NMVOCs by the geographical region where production occurs, similar to the tagging used for primary CO emissions, such that Eqn. 1 becomes

$$\frac{d[CO_{NMVOC,j}]}{dt} = P(CO)_{NMVOC,j} - k[OH][CO_{NMVOC,j}] \tag{4}$$

While the improved method does not allow us to distinguish between production from different precursor NMVOCs (e.g., isoprene vs. monoterpenes) or types of NMVOC emissions (e.g., biogenic vs. anthropogenic), we find that by combining these regional secondary CO tracers with the primary CO tracers we are usually able to infer the likely source of secondary enhancements. An example is discussed in Sect. 5. In both original and improved simulations, secondary CO produced from methane is carried as a single, global tracer.

## 3 Implications for global CO distribution

Figure 1 compares the global distribution of CO from the standard full chemistry simulation to both the original and improved CO-only simulations at the surface and at 500 hPa (the pressure used in the standard GEOS-Chem benchmark and also near the sensitivity maximum for satellites that measure in the thermal infrared). Results shown are for July 2009, but are similar for other months and years (see Supplement, Figs. S1 and S2 for examples). At the surface, the original CO-only model greatly overestimates CO compared to the full chemistry simulation over the continents, with differences of more than 100 ppbv in major biogenic VOC source regions (e.g., Amazon, central Africa, Indonesia, and the Southeast US). This reflects the assumed instantaneous CO production in the surface layer in the original model, whereas in the full chemistry and updated models this production happens more gradually. As a result, the high bias over continents in the original simulation is balanced by a low bias in continental outflow (e.g. the west African plume), at higher altitudes, and in remote regions where there is delayed production following transport.

At 500 hPa, the original CO-only simulation underestimates the full chemistry simulation across much of the globe, again due to delayed CO production during transport. These differences are generally much smaller (4-5 ppbv) than at the surface, reflecting the more widespread nature of secondary production. In a few regions (central Amazon, Indonesia), the original CO-only simulation actually overestimates the full chemistry simulation at 500 hPa by 7-8 ppb. These are regions where high biogenic emissions are coupled with frequent convective activity. In the full chemistry simulation, deep convection rapidly transports biogenic CO precursors to high altitude, and CO production is offset to the middle and upper troposphere (Fisher et al., 2015). In the original CO-only simulation, however, the CO is already present in surface air, and it is CO itself rather than the precursors that is transported to higher altitudes.





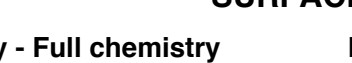

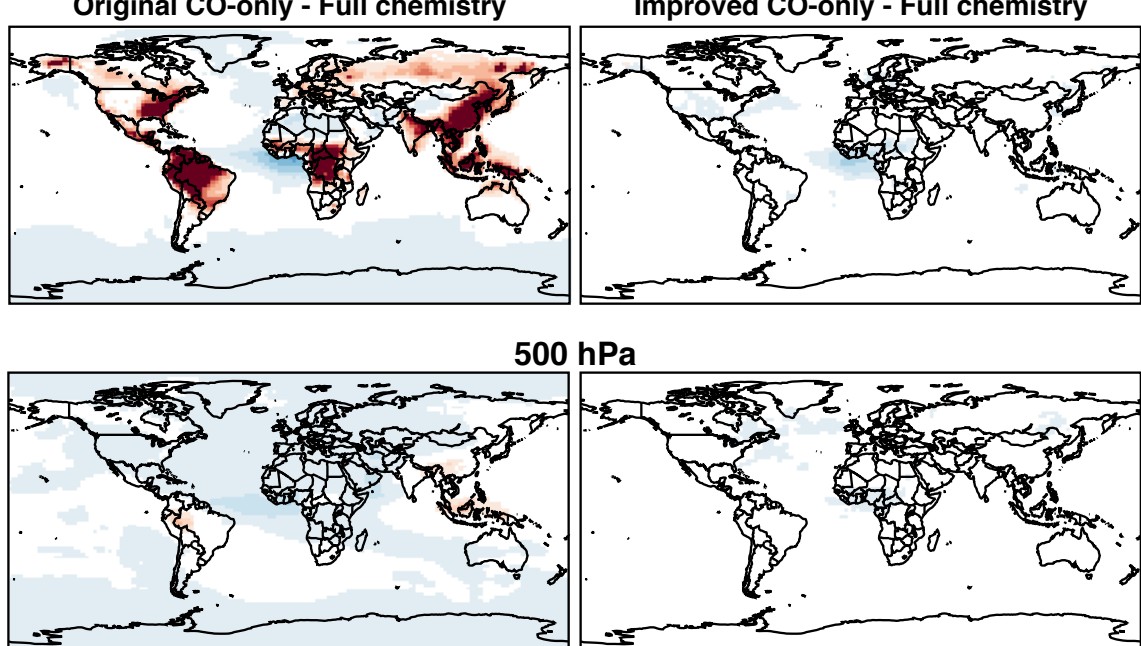

**Figure 1.** July 2009 surface (top) and 500 hPa (bottom) maps of the biases in the CO mixing ratio in the CO-only simulations relative to the full chemistry simulation (used here as reference). Left panels show biases in the original CO-only simulation and right panels show biases in the improved CO-only simulation.

The impacts on the vertical distribution are more clearly seen in Fig. 2, which shows longitudinal cross sections at the equator and at 30°N, and Fig. 3, which shows latitudinal cross sections at 60°W and 0°E. Consistent with the maps in Fig. 1, the cross sections show that the original simulation overestimates CO over continental source regions from the surface to as high as 5 km over South America and Africa, and as high as 10-12 km over Indonesia and China. As seen previously, the high

5    biases over source regions are coupled with low biases in downwind outflow regions.

The right panels of Figs. 1, 2, and 3 show that these issues are largely resolved in the improved CO-only simulation, which simulates less CO in the continental boundary layer and more in the free troposphere and in remote regions downwind of sources. The spatial structure of the changes to the CO distribution are consistent with those seen by Nassar et al. (2010) for carbon dioxide when they introduced a 3-D chemical production source. While some differences remain between the improved

10    CO-only and full chemistry simulations, these are much smaller (less than 6 ppbv everywhere, and less than 3 ppbv outside the African plume) and show a greatly diminished spatial extent. Lingering differences between the two simulations likely reflect



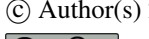

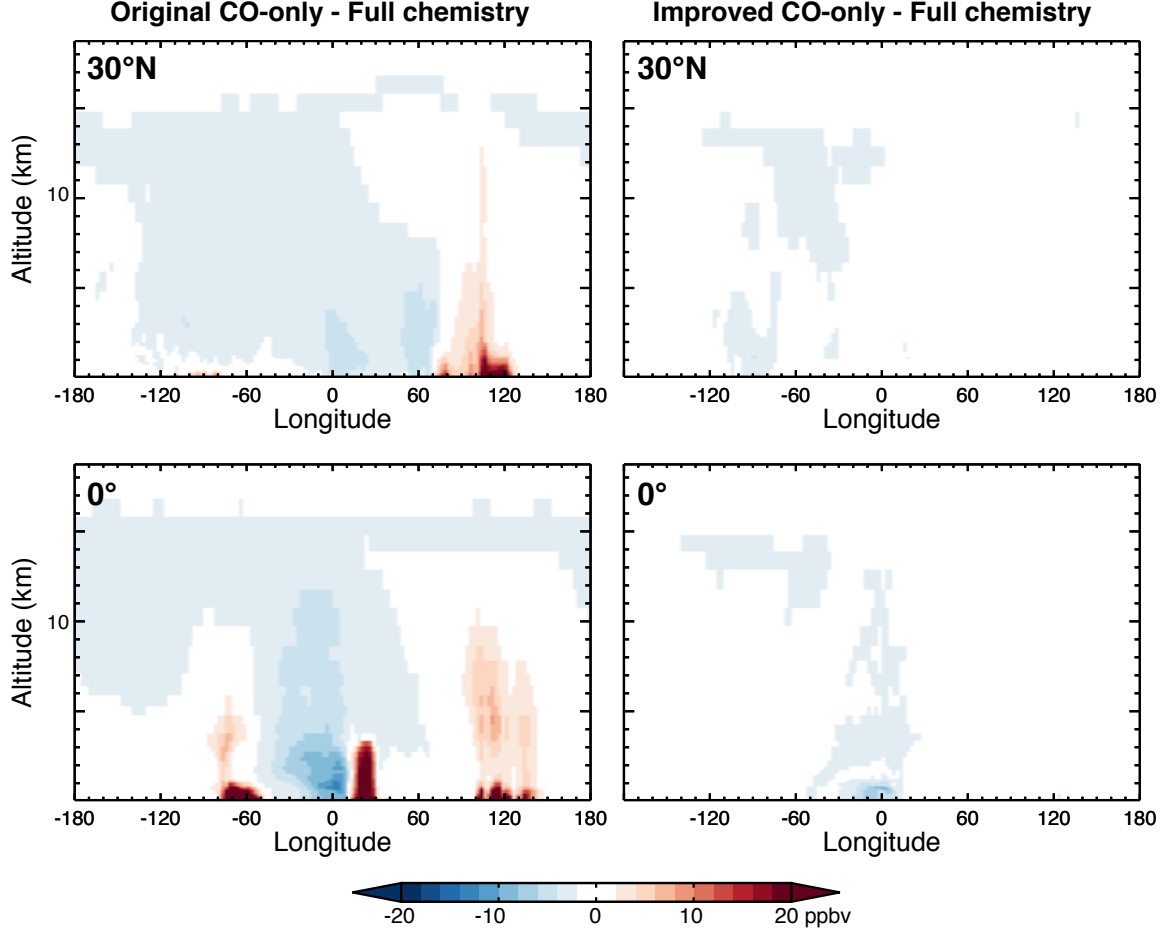

**Figure 2.** July 2009 longitude-altitude cross sections at 30°N (top) and 0° (bottom) of the biases in CO mixing ratio in the CO-only simulations relative to the full chemistry simulation (used here as reference). Left panels show biases in the original CO-only simulation and right panels show biases in the improved CO-only simulation.

use of scaled monthly mean OH concentrations and $P$(CO) fields in the CO-only simulation, which may introduce some offset between the timing and locations of CO production and CO loss.

## 4 Implications for global evaluation with observations

We use a global dataset of ground-based and airborne CO observations to evaluate the improved CO-only simulation in reference to the original simulation and the standard chemistry simulation. As in Sect. 3, we show model output from 2009, which is virtually identical to output from our other simulation years (2010 and 2011). The observations are the same data





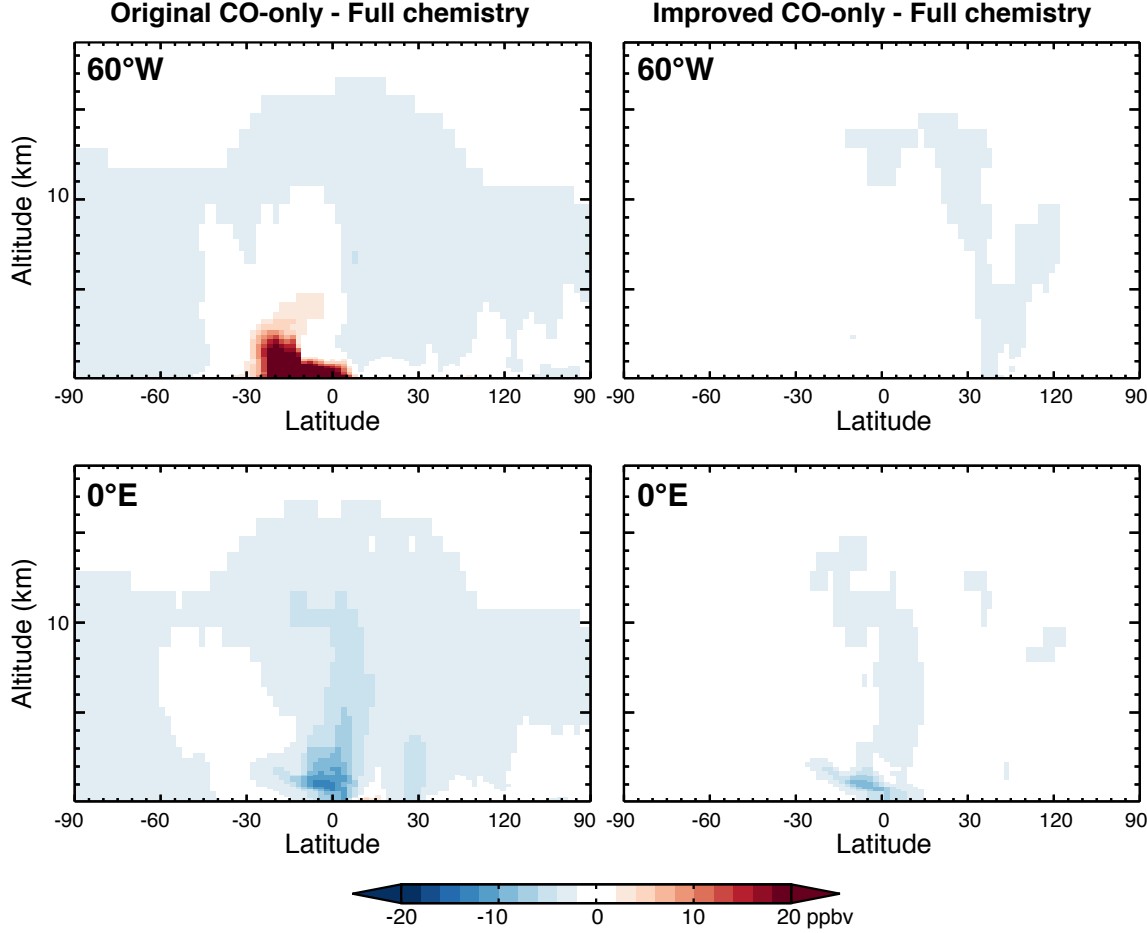

**Figure 3.** July 2009 latitude-altitude cross sections at 60°W (top) and 0°E (bottom) of the biases in CO mixing ratio in the CO-only simulations relative to the full chemistry simulation (used here as reference). Left panels show biases in the original CO-only simulation and right panels show biases in the improved CO-only simulation.

used to benchmark new versions of the full GEOS-Chem model (http://acmg.seas.harvard.edu/geos/geos_benchmark.html), and the software to perform the evaluation is a modified version of the full benchmarking code (https://bitbucket.org/gcst/gc_1yr_benchmark/). A version of the CO-only benchmarking code and relevant observations are available at https://github.com/jennyfisher/CO_Benchmark. The benchmark evaluation produces over 200 plots, of which representative examples are provided in the Supplement (Fig. S3). The full set of plots, including for other simulations years, is available from https://issuu.com/jennyfisher_uow/stacks.



Figure 4 compares the original and updated CO-only and full chemistry simulations to observations at a subset of the sites in the NOAA Global Monitoring Division (GMD) network. The largest differences are seen at the northern mid-latitude continental sites (e.g., up to 20 ppbv at Wendover, Utah, USA; 27 ppbv at Hegyhatsal, Hungary; 47 ppbv at Tae-ahn, Korea), but in all cases these differences are significantly smaller than the observation-model mismatch. On a relative scale, differences

are largest at the Antarctic sites (up to 15%), where they are sufficient to improve (e.g., Palmer) or degrade (e.g., South Pole) agreement with the observations. No sites are located in the major biogenic NMVOC source regions where differences are largest (Fig. 1), and at most sites the surface-level differences between simulations are negligible. This suggests that our current global benchmark dataset is not well situated for evaluating model updates to biogenic emissions and the subsequent chemistry leading to CO production, or for evaluating regional-scale changes to sources (Deng et al., 2014). Expanding the

dataset to include additional monitoring sites in biogenic source regions would improve our ability to constrain future changes in the model.

      The airborne data included in the benchmark dataset tell a similar story – differences between the simulations are generally not large enough to significantly affect the match to the observations (Fig. S4). The largest exceptions, shown in Fig. 5, are over New Guinea and the Amazon – both major biogenic source regions. In both regions, the original simulation significantly

overestimates the observations in surface air. The reduced boundary layer CO concentrations in the improved simulation bring the model into better agreement with the observations. Over New Guinea the bias is reduced but not removed, while over the Amazon the improvement brings the models within (ABLE-2A) or slightly below (ABLE-2B) the variability of the observations. Note that for consistency with benchmarking procedures for the full chemistry simulation, we have included here only the data in the standard GEOS-Chem benchmark (all data available at https://bitbucket.org/gcst/gc_1yr_benchmark/). This

dataset does not include any recent aircraft campaigns, and we therefore do not expect a quantitative match to our simulations (which span 2009-2011), especially near anthropogenic source regions where emissions have changed dramatically over the last few decades (Worden et al., 2013; Yoon and Pozzer, 2014; Strode et al., 2016). The full GEOS-Chem benchmark is similarly conducted for a much more recent year (2013), and updating the benchmark dataset to include more recent surface and airborne observations, including those in regions sensitive to biogenic emissions and chemical production, should be a

priority for the GEOS-Chem community.

      We also evaluated the simulation using quantitative metrics derived from airborne data that describe the CO seasonal cycle (first harmonic) and vertical profile (polynomial terms) in the remote southern hemisphere (see Fisher et al., 2015 for details). The resulting coefficients are given in the Supplement (Figs. S5 and S6). Consistent with the results above, the fit parameters for the full chemistry simulation are more closely approximated by the improved CO-only simulation than the

original version; however, the changes do not significantly impact the match to the observed parameters. Additional southern hemisphere evaluation specific to the Australasian region is provided in the next section.

      Finally, we compared CO total columns from the IASI satellite instrument to those calculated from the original and improved simulations (with a priori and averaging kernel smoothing applied). Due to storage constraints, we limited this analysis to the tropics and southern hemisphere extratropics (70°S to 30°N). Unlike the surface and airborne data, IASI does

sample geographic regions where we see large changes in our simulation; however, the retrieval is mainly sensitive to CO in





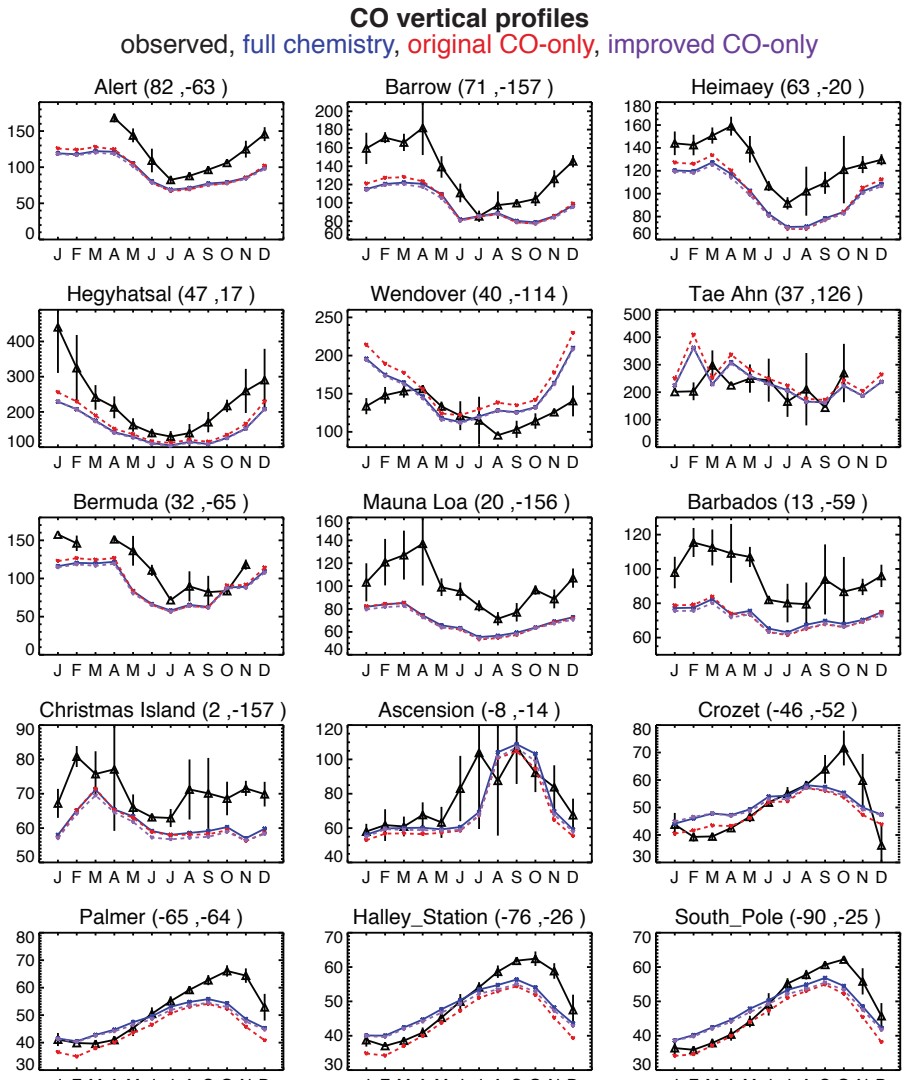

**Figure 4.** Observed and modelled CO seasonal cycle in surface air from a subset of sites in the GEOS-Chem benchmark dataset. Sites are ordered from top left to bottom right by latitude, with coordinates for each given as (°N, °E). Observations are shown in black as multi-year monthly means and standard deviations as provided in the benchmark dataset. Model results are monthly means for 2009 at the location of each site from the full chemistry (solid blue), original CO-only (dashed red), and improved CO-only (dashed purple) simulations. Additional site comparisons can be seen at https://issuu.com/jennyfisher_uow/stacks.

the middle and upper troposphere, where the difference between simulations is generally small (Figs. 1-3). Consistent with the rest of the evaluation, we find that the improved simulation provides a marginally better simulation of the observations, but the





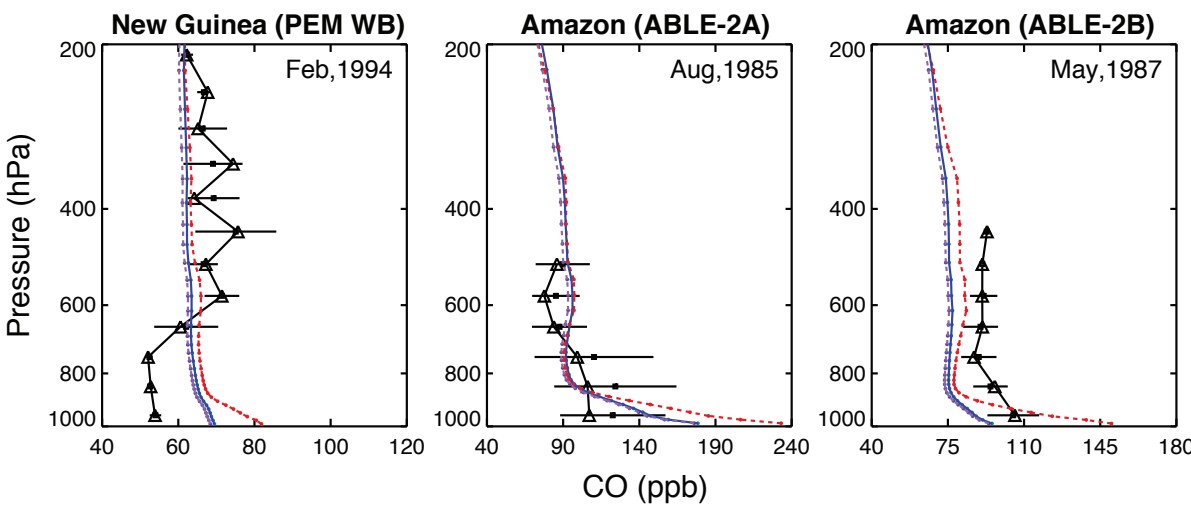

**Figure 5.** Observed and modelled CO vertical profiles from a subset of aircraft campaigns in the GEOS-Chem benchmark dataset. Observations are shown in black with model results for 2009 from the full chemistry (solid blue), original CO-only (dashed red), and improved CO-only (dashed purple) simulations. The month and year during which the aircraft data were collected are provided inset. Additional aircraft comparisons can be seen in Fig. S4 and at https://issuu.com/jennyfisher_uow/stacks.

improvement (~1%) is much smaller than the model bias relative to IASI (~18%) (not shown given the small impact of the simulation change).

## 5 Implications for source attribution: case study for Australasia

Our improved CO-only simulation includes for the first time in GEOS-Chem the ability to geographically tag secondary CO from NMVOC oxidation (see Sect. 2.2.3). Previously, geographic tagging was only available for direct emissions from fossil fuel and biofuel combustion and biomass burning. Outside of the northern hemisphere extratropics, direct emissions are generally responsible for only a small fraction of CO (Duncan et al., 2007; Pfister et al., 2008; Zeng et al., 2015), and so tagging direct emissions only is insufficient for understanding CO or air mass origin. This is especially true in the southern hemisphere tropics and mid-latitudes, where anthropogenic emissions are low and biogenic emissions are large.

To illustrate the new tagging capabilities and their implications, we perform a case study for the Australasian region (Australia and New Zealand). Like much of the southern hemisphere extratropics, Australasia is characterized by very low anthropogenic emissions coupled with episodically high biomass burning emissions in austral spring (October-November; Edwards et al., 2006). Biogenic emissions are also expected to be large in austral summer (December-February) in northern





and southeastern Australia (Guenther et al., 2012; Bauwens et al., 2016), although the magnitude of the biogenic enhancement remains uncertain (Emmerson et al., 2016). As local emissions are generally low, Australasian tropospheric composition is frequently impacted by intercontinental transport of air masses from elsewhere in the southern hemisphere (Gloudemans et al., 2006; Zeng et al., 2012; Buchholz et al., 2016). Significant attention has been paid to transported biomass burning plumes

(e.g., Pak et al., 2003; Gloudemans et al., 2006; Edwards et al., 2006; Zeng et al., 2012). However, recent work by Buchholz et al. (2016) and Té et al. (2016) suggests that NMVOC oxidation dominates over biomass burning as a CO source throughout the year in the Australian extratropics, and in all months except September-October in the tropics. The origin and impact of transport from biogenic source regions in Australasia has not previously been explored.

We evaluate the CO source attribution in Australasia at three locations spanning a range of southern hemisphere en-

vironments: a tropical site at Darwin, Australia (12.4°S, 130.9°E), a mid-latitude site near the Sydney metropolitan area at Wollongong, Australia (34.4°S, 150.9°E), and a remote site with minimal anthropogenic influence at Lauder, New Zealand (45.0°S, 169.7°E). All three sites make total column measurements of CO as part of the Total Carbon Column Observing Network (TCCON, Wunch et al., 2011) and the Network for the Detection of Atmospheric Composition Change (NDACC). Total column CO measurements have previously been compared to the standard GEOS-Chem simulation by Zeng et al. (2015) at all

three sites (using the NDACC record) and by Té et al. (2016) at Wollongong (using both NDACC and TCCON records). Here we use the TCCON data only.

TCCON CO is reported as dry column-average mole fraction, $X_{CO}$, in ppb. We convert this to CO total column, $\Omega_{CO}$, in molecules cm$^{-2}$, using total column O$_2$, $\Omega_{O2}$:

$$\Omega_{CO} = 10^{-9} X_{CO} \times \frac{\Omega_{O2}}{0.2095}. \tag{5}$$

$\Omega_{O2}$ is calculated from variables available in the standard public TCCON files including $p$ (pressure, in hPa) and $X_{H2O}$ (column-averaged water vapour, in ppm), along with an ancillary product created during TCCON data processing called $X_{air}$ (the ratio between the calculated pressure-derived and retrieved dry air columns) as follows:

$$\Omega_{O2} = 0.2095 \cdot 10^{-2} \times \frac{p N_A}{M_{air} g} \times \frac{1}{X_{air} + 10^{-6} X_{H2O}}, \tag{6}$$

where $N_A$ is Avogadro's constant, $M_{air}$ is the molar mass of air in kg mol$^{-1}$, $g$ is the gravitational constant in m s$^{-2}$, and $10^{-2}$

and $10^{-6}$ are unit conversion terms.

Figure 6 compares total column CO from the improved (purple) and original (red) GEOS-Chem CO-only simulations to the TCCON observations (black/gray) from 2009-2011 at the three sites. Consistent with the previous sections, differences between the two CO-only simulations are small. The improved CO-only simulation is in general somewhat higher than the original simulation, especially at Lauder, implying that the CO column is primarily sensitive to transported CO in the free

troposphere rather than local surface sources, as expected. As seen in previous comparisons (Zeng et al., 2015; Té et al., 2016), GEOS-Chem underestimates the total column at all sites, with the best agreement at Lauder.

The CO source attribution from both the original and improved model versions is illustrated in Fig. 7 for 2009. The figure shows three months selected to sample different source influences: February (austral summer), when large biogenic



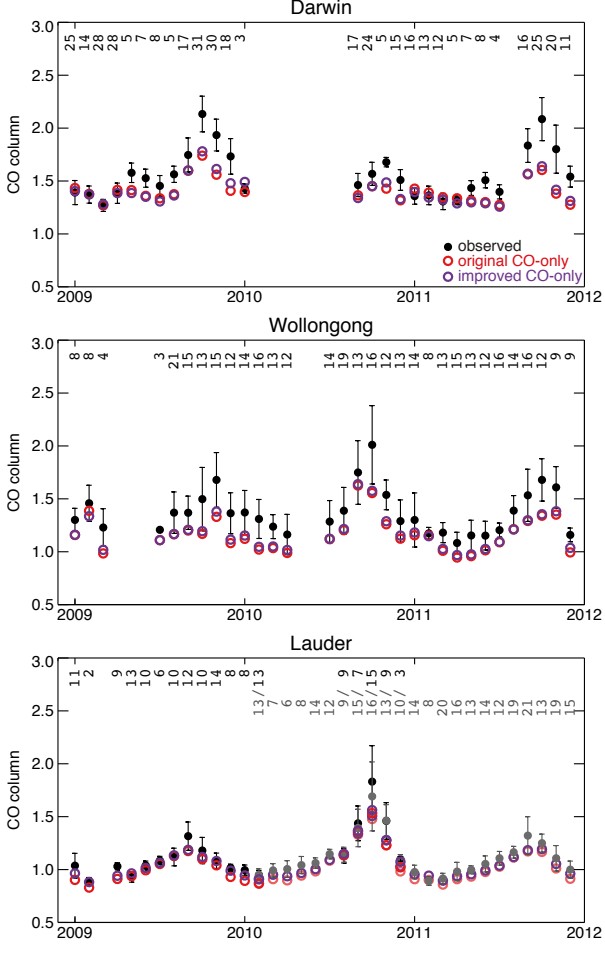

**Figure 6.** Observed and modelled time series of 2009-2011 monthly mean CO total columns in the Australasian region. Observations from the TCCON sites at Darwin (12.4°S, 130.9°E; Griffith et al. 2014a), Wollongong (34.4°S, 150.9°E; Griffith et al. 2014b), and Lauder (45.0°S, 169.7°E; Sherlock et al. 2014a, b) are shown in black as monthly means and standard deviations of all data within each month, with the number of days of data in each month given inset in the figures. Simulated CO from the original CO-only (red) and improved CO-only (purple) has been sampled from the relevant grid boxes only for days with available observations, and smoothed using observed averaging kernels. For Lauder, the spectrometer changed from a Bruker 120HR (dark points/counts) to a Bruker 125HR (light points/counts); both measurements are shown for periods of overlapping data availability.

emissions are expected in northern and southeastern Australia as well as upwind continents; June (austral winter), the start of the dry (burning) season in Darwin but with minimal biogenic and biomass burning emissions elsewhere; and October (austral spring), a period with near-peak biomass burning across much of the southern hemisphere. Contributions are shown from anthropogenic (fossil fuel + biofuel) emissions (purple), biomass burning (orange), and oxidation of NMVOCs (green),





with the remaining CO from methane oxidation (not shown, but equivalent to the difference between 100% and the sum of the stacked contributions). For clarity, the anthropogenic and biomass burning contributions from different regions have been summed globally. The new capability to distinguish regional signatures in the NMVOC source is highlighted for the improved model with different shading.

Figure 7 shows that the contribution from NMVOC oxidation dominates over the emitted primary CO at the mid-latitudes sites, with the two approximately equal at Darwin. In all cases, the improved simulation shows a larger contribution from NMVOC oxidation than in the original simulation. This is in part due to a change in the treatment of NMVOCs from biomass burning and anthropogenic sources (see Sect. 2), which in the original simulation were included with the primary emissions but are now part of the NMVOC contribution. However, the figure also shows that this cannot explain the entirety of the difference,

as the sum of the non-methane contributions (anthropogenic + biomass burning + NMVOC oxidation) is consistently larger in the improved simulation. The difference is particularly large at Lauder, where the influence from primary emissions is generally small. This suggests the improved simulation corrects a consistent underestimate of secondary production found in the original CO-only simulation, with largest impact in remote regions downwind of biogenic sources.

     The source attribution of the NMVOC oxidation term provided in the improved model (Fig. 7, right bars) shows that

the TCCON sites are sensitive to secondary CO produced in a diverse range of environments, including not only Australia but also South America, Africa, and (at Darwin) Indonesia. Biogenic emissions of CO precursors in our simulations are from the MEGAN model, which estimates very high and likely overestimated isoprene across most of northern Australia (Guenther et al., 2012; Bauwens et al., 2016), including the region surrounding the Darwin site. Despite the magnitude of these emissions, we find that typically less than half of the NMVOC source at Darwin is local to Australia. At the mid-latitude sites, transported

CO produced from South American and African NMVOCs provides a large contribution to the column, especially outside of austral summer (>10%). The ability to disaggregate these different contributions can aid in future interpretation of the TCCON data from these sites, including for co-measured species like methane and carbon dioxide.

     More generally, Fig. 8 shows the dominant non-methane contribution to surface and 500 hPa CO over Australasia in February, June, and October. At the surface, oxidation of Australian NMVOCs dominates the CO burden over much of the Aus-

tralian continent, and in February this dominance extends horizontally to New Zealand and vertically to the free troposphere. As noted previously, this is likely due to the large estimated biogenic emissions in austral summer. We see also from Fig. 8 that transported CO produced from South American NMVOC emissions typically dominates the Australasian background (complemented by a similar contribution from African NMVOC oxidation in northern Australia in June). Even in October, at the height of the southern hemisphere burning season, transported chemically produced CO dominates over primary CO (local

or transported) in the free troposphere. We note that there is significant interannual variability in source dominance in October, with a larger contribution from African biomass burning in 2011 and a much larger contribution from South American biomass burning in 2010. In other months, there is limited interannual variability, and the dominant contributions are very similar to those presented here.

     The additional information provided by tagging the NMVOC contribution can aid in the interpretation of both observations

and models, especially in remote regions like much of the southern hemisphere, where the NMVOC source outweighs the





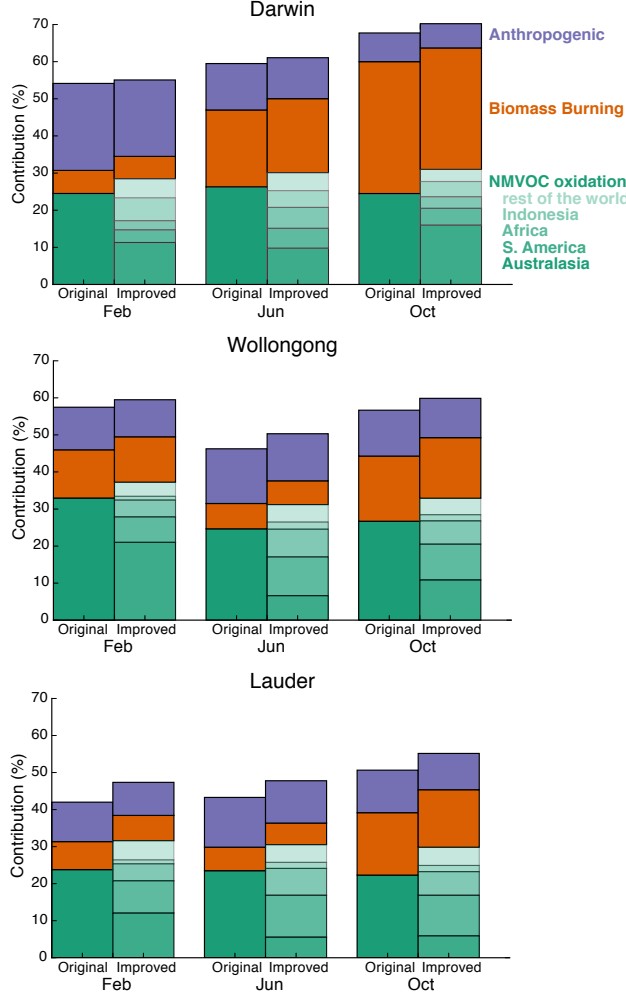

**Figure 7.** Simulated contributions of different sources to total simulated CO at the Australasian TCCON sites in 2009. The figure compares the original (left bars) and improved (right bars) CO-only simulated at the TCCON sites at three times of the year: February (left, austral summer); June (middle, austral winter); and October (right, peak southern hemisphere biomass burning). Colored bars show the percent contributions from anthropogenic emissions (purple), biomass burning emissions (orange), and secondary production from NMVOC oxidation (green). The remainder comes from methane oxidation. For the NMVOC source, the contributions in the improved simulation are divided into source regions (from darkest to lightest): Australasia (112.5-180°E, 48-10°S), South America (120-30°W, 58°S-15°N), Africa (30°W-60°E, 36°S-36°N), Indonesia (95-165°E, 10°S-6°N), and the rest of the world. Note that the region definitions include some ocean areas to capture continental outflow, but the majority of the NMVOC production happens over the continents.

contribution from primary emissions (Zeng et al., 2015). For example, Fisher et al. (2015) found large differences in the ability of four different models to simulate observed CO vertical gradients over the southern ocean. These could not be explained



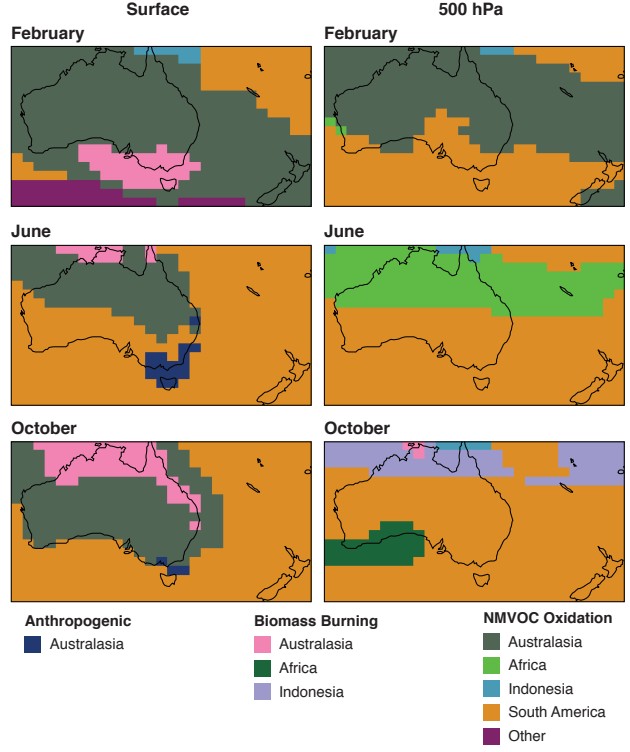

**Figure 8.** GEOS-Chem simulation of the dominant non-methane contribution to Australasian CO at the surface (left) and at 500 hPa (right) in three months, using the improved CO-only simulation with CO source tagged by both type (anthropogenic, biomass burning, or NMVOC oxidation) and region of emissions/production (with regional boundaries the same as in Fig. 7).

by primary CO and were therefore attributed to differences in secondary CO, primarily produced from biogenic NMVOC emissions in South America. However, the authors could not unambiguously deconvolve the effects of differences in chemical production versus transport. Tagging the secondary CO by region of production in the different models would have allowed a quantitative analysis of differences in transport vs. production. Implementation of this capability in multiple models could pave the way for improved interpretation of multi-model comparisons, especially those focused on remote regions.

## 6 Conclusions

We have implemented a major improvement to the representation of secondary CO production in the GEOS-Chem linear "tagged" CO-only simulation, which is frequently used for emission inversion, data interpretation, and chemical forecasting. The improvement targets the production of CO from non-methane volatile organic compounds (NMVOCs), which was previously scaled to NMVOC emissions (assuming fixed yields) and injected into the model surface layer only. This resulted



in a decoupling between the full chemistry and CO-only simulations in both the magnitude and location of secondary CO production. The improved simulation remedies both problems.

In the improved CO-only model, we now use archived CO chemical production rates from the full chemistry simulation, ensuring consistency between the two simulations. We use the methane loss rate (also archived from the full chemistry simulation) to distinguish between CO produced from methane oxidation and that produced from NMVOCs. The latter contribution is for the first time tagged by the geographical region where the production occurs, providing a more comprehensive understanding of air mass origin.

Using the full chemistry simulation as a reference, we showed that the original CO-only simulation greatly overestimates CO in the model surface layer, especially over biogenic source regions (by more than 100 ppbv), due to the assumption of instantaneous surface production. In regions where biogenic emissions coincide with deep convective activity (e.g., South America, Indonesia), the overestimate is expressed throughout much of the troposphere. This reflects the fact that it is typically NMVOCs and/or their intermediate oxidation products that are lifted from the surface layer, with CO production largely happening downwind. The model overestimates at the surface and over major source regions are paired with more diffuse underestimates in outflow regions. Both overestimates and underestimates are largely resolved in the improved CO-only simulation, which shows much closer agreement with the full chemistry.

We compared all three simulations (full chemistry, original CO-only, and improved CO-only) with the global dataset of ground-based and airborne CO observations typically used to benchmark GEOS-Chem. Our comparisons showed that the improvements to the CO-only simulation at the benchmark sites are in most cases significantly smaller than the observation-model mismatch, despite the large differences between simulations seen elsewhere. This inconsistency reflects the geographic representativeness of the data included in the benchmark, with no ground-based sites and few aircraft campaigns that sample air dominantly influenced by secondary production. The current CO benchmark dataset is therefore better placed to monitor the impacts impacts of model updates to primary emissions (anthropogenic and/or biomass burning) than biogenic emissions and their ensuing chemistry. The dataset does not include recent aircraft campaigns targeting biogenic source regions (e.g., GoAmazon, SEAC4RS, SENEX) and the remote atmosphere (e.g., HIPPO, ATom), and updating the benchmark to include these data as well as more recent observations from the surface sites should be a priority for the GEOS-Chem community.

The improved CO-only simulation includes a new capability to geographically tag secondary CO by the region where production occurs. To illustrate this capability, we performed a case study for the Australasian region, where CO is dominated by secondary production. We found that observed total column CO at three TCCON sites across Australasia (Darwin, Wollongong, and Lauder) is sensitive to secondary CO from a range of sources. Throughout much of the year, transported secondary CO dominates over secondary CO produced within Australia, despite large biogenic emissions there, with particularly large contributions from South America followed by Africa and (in the north) Indonesia. Even at the height of the austral biomass burning season that has been the focus of most analysis of intercontinental transport in the southern hemisphere, we show that transported secondary CO from NMVOC oxidation can outweigh transported primary CO from biomass burning.

Linear "tagged" CO-only simulations are used across the atmospheric chemistry community and are of particular value for interpreting field observations and understanding variation in multi-model ensembles. While tagging is generally reserved for





primary emissions, secondary production dominates the CO budget throughout much of the world and especially in the southern hemisphere. Because much of that secondary source comes from oxidation of NMVOCs that are enhanced in biogenic source regions and low elsewhere, there is a geographic signature to secondary CO that can aid in interpretation of observations and model results. We recommend that future attribution studies in regions where primary emissions are low follow the methods
described here to include source attribution of secondary CO contributions.

## 7  Code Availability

The standard GEOS-Chem code is freely accessible to the public by following the guidelines at http://wiki.geos-chem.org/. Updates described here will be included in the standard code once this paper has been accepted, likely after version 11-02.
In the interim, the version 9-01-03 code used here is available at https://github.com/jennyfisher/GEOS-Chem_TaggedCO_v9. 01.03_updated. The provisional version of v11-01 where we have implemented these updates is available by contacting the authors.

The CO-only benchmark code is a condensed and slightly adapted version of the standard GEOS-Chem 1-year benchmark code, available at https://bitbucket.org/gcst/gc_1yr_benchmark/. The CO-only version of the benchmarking code (including the
CO observations used to evaluate the model) is available at https://github.com/jennyfisher/CO_Benchmark.

*Author contributions.*  J.A. Fisher, L.T. Murray, and D.B.A. Jones formulated the method for the improved simulation. J.A. Fisher developed the model code with contributions from L.T. Murray and D.B.A. Jones, performed the simulations, and conducted the analysis. N.M. Deutscher provided the methodology for converting the TCCON data from $X_{CO}$ to $\Omega_{CO}$. J.A. Fisher prepared the manuscript with contributions from all authors.

*Competing interests.*  The authors declare that they have no conflict of interest.

*Acknowledgements.*  This work was funded by a University of Wollongong Vice Chancellor's Postdoctoral Fellowship to JAF with the assistance of resources provided at the NCI National Facility systems at the Australian National University through the National Computational Merit Allocation Scheme supported by the Australian Government. NMD is supported by an ARC-DECRA, DE140100178. TCCON data from Wollongong and Darwin are supported by NASA grants NAG5-12247 and NNG05-GD07G, and Australian Research Council grants
DP140101552, DP110103118, DP0879468 and LP0562346. Technical support for TCCON measurements at Darwin from the Bureau of Meteorology, and formerly the DOE ARM program is gratefully acknowledged. The Lauder TCCON program is core-funded by NIWA through the New Zealand Ministry of Business, Innovation and Employment. We thank Rebecca Buchholz and Clare Paton-Walsh for helpful discussions in the initial stages of this work, and Voltaire Velazco and Dave Pollard for providing the ancillary TCCON variables not available in the public files.





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
