# Peer review of "Improved method for linear carbon monoxide simulation and source attribution in atmospheric chemistry models illustrated using GEOS-Chem v9"

_Geoscientific Model Development, 2017_

## Referee Comment (RC1) · Anonymous Referee #1 · 28 Jun 2017

The manuscript implemented a major improvement to the representation of secondary CO production in the CO-only simulation. The improved CO-only simulation resolves the discrepancy between the full chemistry and CO-only simulations in both the magnitude and spatial distribution. It also includes a new capability of source-region tagging of secondary CO produced from oxidation of non-methane volatile organic compounds. These two improvements help us get a better understanding on CO sources and transport over the remote regions. In general, I found the main points and the structure of this manuscripts are clear. But some of discussions are weak and unclear. Below are

my comments for making the manuscript more concise. I recommended the paper to be published with revision.

Main comments:

Section 4: global evaluation with observations

I understand that the observational data used in this paper is the same data used to benchmark new version of the full GEOS-Chem model. However, it is not proper of using observed climatology to justify model improvements in reference to the original simulation and the standard chemistry simulation. The observed climatology is normally used to do an initial sanity check of simulated mean and seasonality. The authors argue that Figure 4 does not help illustrating the improvements of the new scheme because of the larger observation-model mismatch. To me, the failure of figure 4 is also partly because we are referencing orange to apple. The results would be convincing if the authors compare monthly mean of simulated CO in 2009 to that of GMD surface CO over the same year. What is the measurement frequency for GMD surface CO? The authors should also plot the continuous monthly mean of model output as well as the mean calculated from dates with GMD measurements.

The comparison would probably not change significantly using the observed monthly mean from 2009, but at least that gives us a quantity evaluation of CO simulation in GEOS-Chem full chemistry run and CO-only run.

Please modify Figure 4 as follows:

1) The title should be surface CO instead of CO vertical profiles.

2) The lon and lat of Crozet should be: (-46, 52). Please double check whether this is just a typo in the caption, or extracted from the wrong grid box (-46, -52).

3) It is hard to distinguish the blue and purple lines. please use another color, dark green?

4) Use the monthly mean and standard deviation of GMD surface CO data for 2009 as discussed above.

In addition, I suggest the authors use specific names (instead of original or improved CO-only simulation) for these two versions of CO-only simulation.

Detailed comments:

P4 Line 3: We use the GEOS Chem CTM version 9-01-03 as the based version.

P4 Line 6: Goddard Earth Observing System, Version 5

P6 Line 11: It is not surprise to see the total CO in improved CO-only simulation improves significantly when comparing to the full chemistry CO simulation, since the P(CO) are same in these two runs.

P7, Line 19: China and India also show large overestimate in the surface in the original CO-only run. What causes that?

P7, Line23: For the remote region: dose CO from NMVOC represent 1) CO produced locally through NMVOC transported from source region or 2) transported CO from source region which was produced there or 3) a combination of them?

P7 line 31: similar overestimates seen over Tibetan Plateau, probably related to the deep convection storms in summer time

P11, line 2: largest differences between two CO-only simulations

P11 Lin32: Not sure if it is necessary to discuss the comparison with IASI here.

P12 Figure 4 caption: specify the year range of multi-year monthly mean

P15 Figure 6: For overlapping period, the light purple and dark purple circles look same to me.

P16 Line 7-8: Can you be more specific on the NMVOCs setting in these two version runs? I understand how CO from NMVOCs emitted by biomass burning or anthropogenic emission is in the NMVOCs contribution in the new scheme. But I am confused with the setting of these sources in the old scheme.

P18 Figure 8: adding labels for the locations of three stations in Figure 7.

---

## Referee Comment (RC2) · Anonymous Referee #2 · 30 Jun 2017

The manuscript 'Improved method for linear carbon monoxide simulation and source attribution in atmospheric chemistry models illustrated using GEOS-Chem v9' by Fisher et al. describes an update of the linear CO method which is available in the GEOS-Chem model. Details about the technical implementation (usage of CO production rates from a prior model run instead of yield rates for NMVOC emissions), as well as an comparison of the 'old' and 'new' version with observation data are given. In addition an example of the new source attribution capabilities of the method is presented. As the paper describes an improvement of a current model it is in general

well suited for GMD, before publication, however, major revisions are necessary.

General comments:

The manuscript is in general well written, but especially the description of the model details lacks in clarity. I would suggest to condense the evaluation (more details below) and to describe the model updates as well as the general performance of the model with respect to changes of the CO production rates in more detail.

The authors seem to describe three main updates. The first updates are the improvements described at the end of the Sect. 2.2. These changes are applied in both presented model simulations. Accordingly, the term 'original CO-only simulation' is somewhat misleading as these model results are different from the 'CO-only' method which is the 'status quo' prior to the changes described by the manuscript. The main update are the improvements described in Sect. 2.2.2. The description of these updates seems insufficient to me. For readers which want to implement this method in their own model implementation details are missing. For people of the GEOS-Chem community which might be interested in using the method a user manual or similar (in the supplement) is missing. In addition not all simplifications/assumptions are discussed in detail (see below for details).

In general I would recommend to give proper version numbers to the different updates to make the model changes clearly traceable (e.g. 1.0 original Version, 1.1 updates described in Sect 2.2 etc.). To track the model development it would also be very interesting to show the difference of the model results between 'version 1.0' and 'version 1.1'.

Further I am missing a discussion about how the inter-annual variability of the production and loss rates saved from the full chemistry model influence the results of the 'improved CO only' simulation. Especially as the author state that results from one simulation year are sufficient for the improved model. For interested users it would be very important to see, how critical these pre-calculated production and loss rates are. If dynamics and pre-calculated production rates stem from different years they might not fit to each other and errors/biases are introduced. Please discuss these simplifications. Especially when thinking about sensitivity studies with changed biogenic emissions (e.g. to check the influence of additional source) a new calculation of the secondary CO production (with the full model) seems crucial to me. In this context it would also be interesting to show how large the changes with the new OH fields instead of the OH field used in previous version of the CO-only simulation are.

One problematic thing about the described update is that the amount of needed simulations are doubled. What is the benefit of using first the full model and then the CO-only model? In this context a discussion about the influence of the used CO production rates is important to see if results for different years are dominated by meteorological variability or by variability of the CO production (which of course is also influenced by the meteorological conditions). Especially if users are interest in one specific year it would be interesting to add the source attribution capabilities to the full chemistry model. The source attribution can then be calculated directly during the full model run and the second run (CO only) would not be necessary.

The comparison of the 'original CO' and 'improved CO' to the results of GEOS-Chem using the full chemistry clearly show the improvements of the new version. The authors might discuss shortly that the good agreement between the full chemistry and the 'improved CO only' runs can be expected, as same dynamics, chemical tendencies, emissions and OH-field from the full model run are used by the 'improved

CO only'. Given the small difference between all three model simulations compared to the observations presented in this Section it can be condensed a lot. Of course, a comparison to observation is important. As the differences between all versions are much smaller as the difference between observations and model most of the figures might be moved to the Supplement and the text of this Section can be condensed to the most important findings.

Specific comments:

Sect. 2.1: Please provide a table with the different performed simulations. Instead of 'improved CO only' etc. I would recommend more handy names of the simulations. In general this Section might be moved to the end of Section 2, meaning that first model improvements and at the end the model set-up are discussed.

Sect. 2.2.2: Please clarify which production rates are written out from the full chemistry model run. From my understanding only the total CO production and the methane loss rates are written out, from which the production of CO from NMVOC are then calculated off-line. Meaning you solve the eqn:

$$\frac{dCO}{dt} = E + PCO_{CH4} + PCO_{NMHC} - k[OH][CO] \tag{1}$$

If so, please clearly state the equation you are solving in the description of the improved model.

Figure 1: The figure could be improved a lot by using different color scales for the surface and 500 hPa. Please indicate also the differences in percent in the text.

Section 4: Why are you only evaluating values for the year 2009, while the years 2010 and 2011 are also simulated. Is the difference to the observations in these years similar or different from the values presented for 2009?

The used color coding of Fig. 4, 5 and 6 is problematic. Especially the difference between the blue and the purple color is very hard to see.

P10L1ff: In my opinion it is very problematic to refer to additional graphs (and software) which are available somewhere on the internet. In some years these URLs might not be valid any more. If the graph/software are relevant for the publication then they should be part of the electronic supplement (or get a DOI in any other way).

Figure 4: Please indicate the years for which the observations are averaged. Again, I don't understand why the observations are averaged over many years but the model results not.

Figure 6: The text and symbols are rather small. Are the model results sampled at the same time as the observations or are 'simply' daily average values of the model used when observations are available.

Figure 7: Please clarify - Anthropogenic and biomass burning does only consider primary emissions of CO, right? All production of CO by NMVOC (anthropogenic, biomass burning and biogen) are part of the NMVOC oxidation. If so, please clarify the sentence line P16L9ff.

P19L16ff: Please clarify this paragraph. Do you mean that the improved model should

perform better as the full chemistry model or are you discussing the small difference between the original and the improved model?

Technical comments:

P5L6: Please give a proper reference to the used JPL version.
Figure 4 Title: This figure does not show vertical profiles.

---

## Author Comment (AC1) · 22 Jul 2017

We thank the referee for the useful comments. In what follows, we have included the original referee comments in black, followed by our response in blue, with new text shown in **bold**.

**Anonymous Referee #1**

The manuscript implemented a major improvement to the representation of secondary CO production in the CO-only simulation. The improved CO-only simulation resolves the discrepancy between the full chemistry and CO-only simulations in both the magnitude and spatial distribution. It also includes a new capability of source-region tagging of secondary CO produced from oxidation of non-methane volatile organic compounds. These two improvements help us get a better understanding on CO sources and transport over the remote regions. In general, I found the main points and the structure of this manuscripts are clear. But some of discussions are weak and unclear. Below are my comments for making the manuscript more concise. I recommended the paper to be published with revision.

Main comments:

Section 4: global evaluation with observations

I understand that the observational data used in this paper is the same data used to benchmark new version of the full GEOS-Chem model. However, it is not proper of using observed climatology to justify model improvements in reference to the original simulation and the standard chemistry simulation. The observed climatology is normally used to do an initial sanity check of simulated mean and seasonality. The authors argue that Figure 4 does not help illustrating the improvements of the new scheme because of the larger observation-model mismatch. To me, the failure of figure 4 is also partly because we are referencing orange to apple. The results would be convincing if the authors compare monthly mean of simulated CO in 2009 to that of GMD surface CO over the same year. What is the measurement frequency for GMD surface CO? The authors should also plot the continuous monthly mean of model output as well as the mean calculated from dates with GMD measurements.

The comparison would probably not change significantly using the observed monthly mean from 2009, but at least that gives us a quantity evaluation of CO simulation in GEOS-Chem full chemistry run and CO-only run.

We appreciate the suggestion. In digging through the standard benchmark dataset, we realised that the surface CO data included were not multi-year means as expected but rather monthly means for 2005 only. We now follow the reviewer's suggestion and compare the simulation to the observed data from the same year (e.g., 2009 in Fig. 4). The agreement does improve, although the message remains the same: the observation-model mismatch is larger than the differences between the three simulations. Due to different data availability in 2009, we replaced two sites shown in the original Fig. 4 (Bermuda and Palmer) with nearby sites (Key Biscayne and Syowa, respectively).

Figure 4 now shows the 2009 monthly means for the model. We only saved the global model output at monthly time resolution, so we are unable to also plot the mean calculated only from dates with GMD measurements.

We have also used this opportunity to update the CO data used by the GEOS-Chem Support Team for the standard full chemistry benchmarking, which uses model output from 2013.

Please modify Figure 4 as follows:

1) The title should be surface CO instead of CO vertical profiles.
Fixed.

2) The lon and lat of Crozet should be: (-46, 52). Please double check whether this is just a typo in the caption, or extracted from the wrong grid box (-46, -52).
Fixed. This was in fact extracted from the wrong gridbox – and appears to have been for all earlier versions of the GEOS-Chem benchmark as well. This has now been fixed in Figure 4 and reported to the GEOS-Chem Support Team to fix in the public benchmark as well.

3) It is hard to distinguish the blue and purple lines. please use another color, dark green?
It is a challenge to find colors that are distinguishable both in general and for a red-green colorblind reader. We have replaced the solid blue line with solid orange in Figs. 4 & 5. While the orange and red lines may look similar to some colorblind readers, the dashing will help distinguish these, and the lines are now more easily distinguished for the non-colorblind reader.

4) Use the monthly mean and standard deviation of GMD surface CO data for 2009 as discussed above.
Fixed as discussed above.

In addition, I suggest the authors use specific names (instead of original or improved CO-only simulation) for these two versions of CO-only simulation.
In response to a comment from Referee #2, we changed the name "original" to "base". We use the terms "base" and "improved" to avoid any ambiguity in the figures and text. Because the GEOS-Chem model itself (of which the CO-only simulation is one part) already has version numbers, we feel it would be confusing to replace the names with a version. In the absence of short names that could clearly and succinctly encapsulate the differences between the two versions of the simulation, we prefer to retain "base" and "improved", which we use consistently throughout the text and figures. We would be happy to consider any specific suggestions for clearer version names. To further improve the clarity of these names for the readers, we now include information about the differences between the two in the figure captions for Fig. 1 (first model-model comparison figure) and Fig. 4 (first model-observation comparison figure):

"**The main difference between the two CO-only simulations is the vertical distribution of the CO source from NMVOCs, which is 3-dimensional in the improved simulation but surface-only in the base simulation.**"

Detailed comments:

P4 Line 3: We use the GEOS Chem CTM version 9-01-03 as the based version.
Fixed: "We use the GEOS-Chem CTM version 9-01-03 **as the base version**…"

P4 Line 6: Goddard Earth Observing System, Version 5

Fixed: "Goddard Earth Observing System**, version 5** (GEOS-5)."

P6 Line 11: It is not surprise to see the total CO in improved CO-only simulation improves significantly when comparing to the full chemistry CO simulation, since the P(CO) are same in these two runs.

Yes precisely – the goal is to quantify how large the discrepancy was in the original (standard) simulation. We have added the following to the text when we first discuss the improved agreement with the full chemistry simulation:

P8 line 33-P9 line 5: "While some differences remain between the improved CO-only and full chemistry simulations, these are much smaller (less than 6 ppbv / 5% everywhere, and less than 3 ppbv /3% outside the African plume) and show a greatly diminished spatial extent. **The improved agreement is expected, as the spatial distribution of CO production is now the same in the two simulations.** Lingering differences likely reflect use of scaled monthly mean OH concentrations and $P$(CO) fields in the CO-only simulation, which may introduce some offset between the timing and locations of CO production and CO loss."

P7, Line 19: China and India also show large overestimate in the surface in the original CO-only run. What causes that?

Although it's not visible on the color scale used in the figure, these overestimates are smaller. They come from anthropogenic and/or biomass burning VOCs. We now add this to the text:

P8, lines 9-14: "At the surface, the base CO-only model greatly overestimates CO compared to the full chemistry simulation over the continents, with differences of more than 100 ppbv (60-80%) in major biogenic VOC source regions (e.g., Amazon, central Africa, Indonesia, and the Southeast US). **Similar, smaller (up to 50 ppbv / 10-15%) effects are present in regions with elevated VOC emissions from anthropogenic and biomass burning sources (e.g., China, Alaska). These overestimates** reflect the assumed instantaneous CO production from VOCs in the surface layer in the base model, whereas in the full chemistry and updated models this production happens more gradually."

P7, Line 23: For the remote region: dose CO from NMVOC represent 1) CO produced locally through NMVOC transported from source region or 2) transported CO from source region which was produced there or 3) a combination of them?

We are unsure what text the referee was referring to, as P7, line 23 does not discuss CO from NMVOC. We now clarify this issue earlier, after Equation 5:

P7, lines 28-30: "**In other words, $CO_{NMVOC,j}$ from region $j$ represents the CO that was produced within the boundaries of region $j$, regardless of the origin of the NMVOC (i.e., local vs. transported). As most precursor NMVOCs have short atmospheric lifetimes, most $CO_{NMVOC,j}$ will derive from NMVOCs emitted in region $j$.**"

P7 line 31: similar overestimates seen over Tibetan Plateau, probably related to the deep convection storms in summer time

We have added this region to the discussion. In the improved version of the figure, the overestimate appears centred over southern China. Precursors in this region are likely to include anthropogenic emissions, and so we have modified the accompanying text accordingly:

P8 lines 19-21: "In a few regions (central Amazon, Indonesia, **and southern China**), the base

CO-only simulation actually overestimates the full chemistry simulation at 500 hPa by 7-8 ppb (10-12%). These are regions where **large NMVOC sources** are coupled with frequent deep convective activity."

P11, line 2: largest differences between two CO-only simulations
Fixed: "The largest differences **between the two CO-only simulations** are seen…"

P11 Lin32: Not sure if it is necessary to discuss the comparison with IASI here.
We disagree – while we found that the impact was small, one of the motivations for the work was the possibility that the original treatment of secondary CO would degrade comparisons with satellite instruments that are most sensitive in the mid-troposphere (such as IASI). We prefer to include this text as it may be relevant to other readers who have the same concern.

P12 Figure 4 caption: specify the year range of multi-year monthly mean
See earlier comments – we now instead show the 2009 data and specify this in the caption.

P15 Figure 6: For overlapping period, the light purple and dark purple circles look same to me.
Fixed

P16 Line 7-8: Can you be more specific on the NMVOCs setting in these two version runs? I understand how CO from NMVOCs emitted by biomass burning or anthropogenic emission is in the NMVOCs contribution in the new scheme. But I am confused with the setting of these sources in the old scheme.
This is explained in Section 2.2.1, and we now also re-iterate it here:
Pg 17, lines 9-11: "**In the base simulation, these** were included with the primary emissions **as they were calculated by increasing primary anthropogenic and biomass burning emissions by 19% and 11%, respectively (Sect. 2.2.1); whereas in the improved simulation**, they are part of the NMVOC contribution."

P18 Figure 8: adding labels for the locations of three stations in Figure 7.
Fixed

---

## Author Comment (AC2) · 22 Jul 2017

We thank the referee for the useful comments. In what follows, we have included the original referee comments in black, followed by our response in blue, with new text shown in **bold**.

**Anonymous Referee #2**

The manuscript 'Improved method for linear carbon monoxide simulation and source attribution in atmospheric chemistry models illustrated using GEOS-Chem v9' by Fisher et al. describes an update of the linear CO method which is available in the GEOS-Chem model. Details about the technical implementation (usage of CO production rates from a prior model run instead of yield rates for NMVOC emissions), as well as an comparison of the 'old' and 'new' version with observation data are given. In addition an example of the new source attribution capabilities of the method is presented. As the paper describes an improvement of a current model it is in general well suited for GMD, before publication, however, major revisions are necessary.

General comments:

The manuscript is in general well written, but especially the description of the model details lacks in clarity. I would suggest to condense the evaluation (more details below) and to describe the model updates as well as the general performance of the model with respect to changes of the CO production rates in more detail.

The authors seem to describe three main updates. The first updates are the improvements described at the end of the Sect. 2.2. These changes are applied in both presented model simulations. Accordingly, the term 'original CO-only simulation' is somewhat misleading as these model results are different from the 'CO-only' method which is the 'status quo' prior to the changes described by the manuscript.
We have clarified the text in Section 2.2 to explain that these updates are effectively bug fixes and that these are part of the current standard ('status quo') GEOS-Chem CO-only simulation (v11-01). The only non-standard change described in this section is the OH fields, discussed below. We are sensitive to the concern that the term "original" is misleading, and have replaced in with "base" throughout the manuscript (except in the abstract, where we feel it would be confusing without additional context).

The paragraph in question now reads:
P5, lines 25-32: "**We also fixed a few minor problems in the v9-01-03 CO-only simulation to bring it up-to-date with the current public version (v11-01) and to make it more compatible with the full chemistry simulation. Specifically**, we implemented optional non-local mixing in the planetary boundary layer (Lin and McElroy, 2010) and the centralized chemistry timestep (http: //wiki.seas.harvard.edu/geos-chem/index.php/Centralized_chemistry_time_step) – both of which are defaults in the v9-01-03 standard full chemistry simulation but were missing from the CO-only simulation. We also added diurnal scaling of the monthly mean [OH] fields based on the cosine of the solar zenith angle. This is the same method used in all other offline simulations in GEOS-Chem, and is an objective improvement compared to using a single monthly mean value in each grid box. All of these updates are **included in both the base and improved versions compared below.**"

The main update are the improvements described in Sect. 2.2.2. The description of these updates seems insufficient to me. For readers which want to implement this method in their own model implementation details are missing. For people of the GEOS-Chem community which might be interested in using the method a user manual or similar (in the supplement) is missing. In addition not all simplifications/assumptions are discussed in detail (see below for details).

We have addressed all the specific comments and suggestions below, and hope that this now covers all previously missing implementation details. We have also expanded the Section 7 on Code Availability to more clearly explain how users in the GEOS-Chem community can apply the changes:

P21, lines 14-19: "The standard GEOS-Chem code is freely accessible to the public by following the guidelines at http://wiki.geos-chem.org/. Updates described here will be included in the standard code once this paper has been accepted, likely after version 11-02. In the interim, the version 9-01-03 code used here is available at https://github.com/jennyfisher/GEOS-Chem_TaggedCO_v9.01. 03_updated. **The standard GEOS-Chem code is distributed as a git repository, and the code updates described here can be added to a standard v9-01-03 code repository by pulling the branch linked above.** The provisional version of v11-01 where we have implemented these updates is available by contacting the authors."

We have also added a link to the user manual, which provides all details for compiling and running GEOS-Chem (none of which have changed in this version):

P21, lines 19-20: "**Instructions for running the GEOS-Chem model, including the CO-only simulation, are provided in the GEOS-Chem User's Guide: http://acmg.seas.harvard.edu/geos/doc/man/.**"

In general I would recommend to give proper version numbers to the different updates to make the model changes clearly traceable (e.g. 1.0 original Version, 1.1 updates described in Sect 2.2 etc.).

Because the GEOS-Chem model itself (of which the CO-only simulation is one part) already has version numbers, we feel it would be confusing to replace the names with a version. In the absence of short names that could clearly and succinctly encapsulate the differences between the two versions of the simulation, we prefer to retain simulation names "base" and "improved", which we use consistently throughout the text and figures.

To track the model development it would also be very interesting to show the difference of the model results between 'version 1.0' and 'version 1.1'.

As described above, most of the minor changes described in Section 2.2 were essentially bug fixes (already made in the public version 11-01), and we have updated the text to clarify this. The only non-standard change described in this section is the OH fields. As described below, we have added an extra figure to the Supplement (new Fig. S1) to show the magnitude of the changes due to OH (in essence, comparing 'v1.0' and 'v1.1').

Further I am missing a discussion about how the inter-annual variability of the production and loss rates saved from the full chemistry model influence the results of the 'improved CO only' simulation. Especially as the author state that results from one simulation year are sufficient for the improved model. For interested users it would be very important to see, how critical these pre-calculated production and loss rates are. If dynamics and pre-calculated production rates stem from different years they might not fit to each other and errors/biases are introduced. Please discuss these simplifications. Especially when thinking about sensitivity studies with changed biogenic emissions (e.g. to check the influence of additional source) a new calculation of the secondary CO production (with the full model) seems crucial to me.

This is an important point. We have added to the Supplement a section (new Section S1) and a figure (new Fig. S2) to explain that the interannual variability in the improved CO-only simulation is dominated by meteorology and emissions, with only a very minor contribution from the pre-calculated chemical production rates. We agree with the referee that some specific applications (e.g., evaluating differences in biogenic emissions) would require the user to re-calculate the secondary production fields, and this is why we originally noted in the text that an interested user could easily run the standard chemistry simulation as required. We now explicitly provide examples of when this step may be necessary. The updated text now reads:

P6, lines 23-29: "For many applications, use of a single model year (e.g., that used in the 1-year benchmarking process) will be sufficient—**as discussed in the Supplement, interannual variability in simulated CO is dominated by variability in the meteorology and primary emissions with only a very minor contribution from variability in the CO production rates (Sect. S1, Fig. S2)**. However, the code is designed in such a way that an interested user could easily re-run the standard chemistry simulation to save the CO production fields for a specific year of interest. **This extra step would generally only be necessary for changes likely to significantly impact chemical production (for example, different biogenic emissions or meteorological fields)**."

We would also note that in the past, the CO-only simulation has been applied to a broad range of problems (see for example references in the introduction), and for many of these the small errors introduced by using a single year for the chemical production will not be important.

In this context it would also be interesting to show how large the changes with the new OH fields instead of the OH field used in previous version of the CO-only simulation are.

We have re-written this paragraph to add detail, and have added an extra figure to the Supplement (new Fig. S1) to show the magnitude of the changes due to OH:

P5, lines 15-21: "Previously, the CO-only simulation used OH from earlier versions of the model, **typically v5-07-08** (e.g., Kopacz et al., 2010; Fisher et al., 2010; Jiang et al., 2011, **2017**), to mitigate a known OH high bias in more recent versions of GEOS-Chem. Here, we use OH from v9-01-03 in both our base and improved CO-only simulations to maintain consistency with the full chemistry model and to ensure all changes are due to the new representation of chemical production rather than differences in OH. **Global annual mean OH is about 5% larger in v9-01-03 (11.4·$10^5$ molecules cm$^{-3}$; http://wiki.seas.harvard. edu/geos-chem/index.php/Mean_OH_concentration) than in the default v5-07-08 (10.8·$10^5$ molecules**

**cm⁻³; Evans and Jacob, 2005; Kopacz et al., 2010). As a result, CO is systematically lower when using the updated OH fields, as shown in Fig. S1."**

One problematic thing about the described update is that the amount of needed simulations are doubled. What is the benefit of using first the full model and then the CO-only model? In this context a discussion about the influence of the used CO production rates is important to see if results for different years are dominated by meteorological variability or by variability of the CO production (which of course is also influenced by the meteorological conditions). Especially if users are interest in one specific year it would be interesting to add the source attribution capabilities to the full chemistry model. The source attribution can then be calculated directly during the full model run and the second run (CO only) would not be necessary.

As described in the comment above, we now explicitly show that the interannual variability in simulated CO is dominated by meteorology and emissions, with only a very minor contribution from secondary CO production. This means that many users who are not specifically focusing on e.g. biogenic emissions or biogenic source regions will still be able to run only the CO-only simulation. Relative to the full chemistry simulation, the CO-only simulation is 6 times faster and requires only ~40% as much memory, so there are significant advantages to the GEOS-Chem community to maintaining a CO-only simulation (e.g., for use in forecasting, source attribution, interpreting multi-model ensembles, etc.). This is particularly useful for the four-dimensional variational data assimilation scheme in GEOS-Chem, which can require 20-40 iterations of the model simulation when inverse modeling CO emissions. We agree that adding CO source attribution capabilities to the full chemistry model would also be a useful future addition (that could be maintained in parallel to the CO-only simulation) – but that would require additional code development that is outside the scope of this work.

The comparison of the 'original CO' and 'improved CO' to the results of GEOS-Chem using the full chemistry clearly show the improvements of the new version. The authors might discuss shortly that the good agreement between the full chemistry and the 'improved CO only' runs can be expected, as same dynamics, chemical tendencies, emissions and OH-field from the full model run are used by the 'improved CO only'.

We now clarify that the good agreement between the full chemistry and improved simulations is expected (see also response to Referee #1):

P9, lines 2-3: **"The improved agreement is expected, as the spatial distribution of CO production is now the same in the two simulations."**

Given the small difference between all three model simulations compared to the observations presented in this Section it can be condensed a lot. Of course, a comparison to observation is important. As the differences between all versions are much smaller as the difference between observations and model most of the figures might be moved to the Supplement and the text of this Section can be condensed to the most important findings.

We believe it is important to give the readers a thorough (and visual) understanding of the

comparison to observations. From Figs. 1-3, one would expect to see a large change in the model-observation agreement between the old and new simulations. The fact that this isn't the case (shown in Fig. 4) is an important result from this work – showing that the GEOS-Chem benchmark data are not sufficiently representative to constrain current model uncertainties. The aircraft are useful for showing the magnitude of the change in regions with enhanced biogenic emissions that are not captured in the surface data.

We feel this section will be of interest to some readers and is already condensed, with only 4 paragraphs of text and all but two figures in the Supplement. We therefore prefer to leave Section 4 in place.

Specific comments:

Sect. 2.1: Please provide a table with the different performed simulations.
We now include a table with the simulation details (P4, new Table 1).

Instead of 'improved CO only' etc. I would recommend more handy names of the simulations.
We have not been able to come up with a better succinct name that clearly communicates the updates made to this simulation, but we would be happy to consider any specific suggestions.

In general this Section might be moved to the end of Section 2, meaning that first model improvements and at the end the model set-up are discussed.
We prefer to retain the order as-is as it allows the text to flow directly from the description of the new simulation to the implications of the changes. We feel this flow would be lost by moving Section 2.1 to the end of Section 2.

Sect. 2.2.2: Please clarify which production rates are written out from the full chemistry model run. From my understanding only the total CO production and the methane loss rates are written out, from which the production of CO from NMVOC are then calculated off-line.
We now clarify which rates are saved from the full chemistry model and that the methane and NMVOC production are calculated offline.
P6, lines 30-31: "The total $P(CO)$ **saved from the full chemistry model** is split **offline** into contributions from methane ($P(CO)_{CH4}$) and from NMVOCs ($P(CO)_{NMVOC}$) by also saving the methane loss rates ($L(CH4)$) from the full chemistry simulation."

Meaning you solve the eqn:

$$dCO/dt = E + PCO_{CH4} + PCO_{NMHC} - k[OH][CO] \ (1)$$

If so, please clearly state the equation you are solving in the description of the improved model.
We have also added a new Equation (2) that shows the equation solved in the improved model.
P7, line 9: $d[CO]/dt = E + P(CO)_{CH4} + P(CO)_{NMVOC} - k[OH][CO]$        **(2)**

Figure 1: The figure could be improved a lot by using different color scales for the surface and 500 hPa.

Fixed – we now use a color scale of -50 to 50 ppb for surface CO and -10 to 10 ppb for 500 hPa CO.

Please indicate also the differences in percent in the text.
We now also state the differences in percent in the text everywhere differences are quantified.

Section 4: Why are you only evaluating values for the year 2009, while the years 2010 and 2011 are also simulated. Is the difference to the observations in these years similar or different from the values presented for 2009?
As we state at the start of Section 4, results from the three different years are indistinguishable in terms of model-observation comparisons. See P9, lines 7-8: "As in Sect. 3, we show model output from 2009, which is virtually identical to output from our other simulation years (2010 and 2011)."

The used color coding of Fig. 4, 5 and 6 is problematic. Especially the difference between the blue and the purple color is very hard to see.
Fixed – see response to Referee #1 – blue has been changed to orange.

P10L1ff: In my opinion it is very problematic to refer to additional graphs (and software) which are available somewhere on the internet. In some years these URLs might not be valid any more. If the graph/software are relevant for the publication then they should be part of the electronic supplement (or get a DOI in any other way).
We were hesitant to make the Supplement unreadably long (each yearly benchmark is 41 pages), and so chose to post the extra benchmark plots at a large online publishing company unlikely to disappear. Nonetheless, we are sensitive to the referee's concern and will include all 3 benchmarks in the Supplement of the revised version.

Figure 4: Please indicate the years for which the observations are averaged. Again, I don't understand why the observations are averaged over many years but the model results not.
Fixed – see response to Referee #1. Figure 4 now shows 2009 results for both model and observations.

Figure 6: The text and symbols are rather small.
We have now magnified the figure as much as can be done in the GMDD latex template (making the figure any larger within the template causes it and all subsequent figures to appear after the references). We expect a larger version of the figure to appear in the final version.

Are the model results sampled at the same time as the observations or are 'simply' daily average values of the model used when observations are available.
The model results are daily mean values for days when observations are available, as we now state in the figure caption:
P16, Fig. 6 caption: "Simulated **daily mean** CO from the base CO-only (red) and improved CO-only (purple) has been sampled from the relevant grid boxes only for days with available observations…"

Figure 7: Please clarify - Anthropogenic and biomass burning does only consider primary emissions of CO, right? All production of CO by NMVOC (anthropogenic, biomass burning and

biogen) are part of the NMVOC oxidation. If so, please clarify the sentence line P16L9ff.

As explained in Section 2.2.1 and now re-iterated here, in the base model secondary production from anthropogenic and biomass burning NMVOC was included as part of the primary source. We now repeat this information here:

Pg 17, lines 9-11: "In the base simulation, these were included with the primary emissions **as they were calculated by increasing primary anthropogenic and biomass burning emissions by 19% and 11%, respectively (Sect. 2.2.1); whereas in the improved simulation**, they are part of the NMVOC contribution."

P19L16ff: Please clarify this paragraph. Do you mean that the improved model should perform better as the full chemistry model or are you discussing the small difference between the original and the improved model?

We have reorganised this text to explicitly state that the differences between the simulations are smaller than the observation-model mismatch:

P20, lines 22-24: "Our comparisons showed that at the benchmark sites, **the differences between the base and improved CO-only simulations** are in most cases significantly smaller than the observation-model mismatch, despite the large differences between **the** simulations seen elsewhere."

Technical comments:

P5L6: Please give a proper reference to the used JPL version.
Added.

Figure 4 Title: This figure does not show vertical profiles.
Fixed

---

## Author Response (AR2)

We thank the referees for the careful re-read of our revised manuscript. In what follows, we have included the original comments in black, followed by our response in blue, with new text shown in **bold**. On page 3, we list all relevant changes made in the manuscript and supplement.

**Report #1 - Anonymous Referee #2**

In general the authors considered all reviewer comments. In particular, the changed simulations names, inclusion of the new Table 1 and the updated figures help a lot to improve the manuscript. However, I think that some points could have been described in more detail or more clearly (e.g. the new equation 2). This would increase the quality of the document.
I overall agree with the points where the authors did not incorporate the suggested changes.

The issue of the influence of the prescribed CO production rates, however, is still not discussed in sufficient detail. The authors added a Section in the supplement clearly showing that for the years 2009/2010/2011 the variability of meteorology/emissions influence CO more as the variability of the secondary CO production. However, comparing only 2009/2010/2011 this could be by chance. As the usage of the prescribed CO production rates is crucial for a good performance of the linear CO only model this needs to be discussed in more detail - e.g. what happens under 'extreme' conditions (e.g. strong biomass burning events). Especially, as these CO production rates are tagged to distinguish the geographical regions, the influence of the prescribed production rates on the tagging results should be investigated.

We have now strengthened the discussion in the Supplement (Section S1) to address both of the referee's concerns.

First, we now point out that our simulation years include strong biomass burning events and are therefore appropriate for testing the influence of variability in $P(CO)$ even under extreme conditions:

"Figure S2 shows the results from these comparisons for July (other months are similar). At both the surface and 500 hPa, the interannual variability of $P(CO)$ makes only a minor contribution (<7%) to the overall variability. Other factors (meteorology, emissions) clearly dominate the simulated year-to-year differences. **The impacts of large, yearly-varying biomass burning events can clearly be seen over Indonesia (intense fires in 2009, no fires in 2010), central Canada (intense fires in 2010, no fires in 2009), and western Russia (intense fires in 2010, weak fires in 2009). These year-to-year differences in biomass burning drive large interannual variability in total CO (Fig. S2a), but the majority of this is driven by the variability in primary emissions (Fig. S2b) rather than the variability in $P(CO)$ (Fig. S2c).** We performed the same test for 2011 and found the same result."

Second, we added a figure (Fig. S3) and discussion in Section S1 regarding the influence of $P(CO)$ variability on source attribution:

"**We also evaluated the impacts for CO source attribution using the same simulations (2009, 2009 meteorology/emissions with 2010 $P(CO)$, and 2010). For each simulation, we**

**calculated the contributions from different source types (primary fossil anthropogenic, primary biomass burning, and NMVOC oxidation) and source regions (for NMVOC oxidation) at the three Australasian TCCON sites described in Sect. 5 of the main text. Figure S3 shows the results for February, when there were extremely large fires in southeast Australia in 2009 but none in 2010, and October, when there were much larger fires in northern Australia in 2009 than 2010 and larger fires in southern Africa in 2010 than in 2009.**

**Figure S3 shows that meteorology and primary emissions generally dominate the interannual variability in source contributions. The largest year-to-year differences at all sites come from primary biomass burning emissions. Interannual variability in $P$(CO) can also drive small differences in CO from NMVOC oxidation, particularly near large and variable emission sources. This is most evident at Darwin in October, when the large decrease in Australian biomass burning emissions from 2009-2010 was associated with a decrease in $P$(CO) from Australian NMVOC oxidation as well. Downwind of sources, these impacts become negligible. For example, the larger burning in southern Africa in 2010 than 2009 can clearly be seen in the primary emissions contribution at Wollongong and Lauder, but the changes in the African NMVOC source are minimal at both sites."**

We further clarified the summary paragraph of Section S1:

"These tests suggest that use of a single year of $P$(CO) fields will be appropriate for many applications of the CO-only simulation. The errors introduced by this approach are significantly smaller than the difference between the base and improved methods described here. The exception would be any studies specifically focused on secondary CO (for example, evaluating impacts of using different biogenic emission inventories)**, or specifically evaluating source attribution near major and variable primary emission sources**. In these cases, it would be necessary for an interested user to re-run the full chemistry simulation and save $P$(CO) for each scenario to be evaluated."

Finally, we now refer to this evaluation in Section 5 of the main text:

P17 L5: "The sensitivity of the source attribution to interannual variability in $P$(CO) is shown in Fig. S3 in the Supplement."

Minor comments:
On p5 l 25 of the revised manuscript you are now writing: ".... to bring them up to date with". Does this mean that these updates were already included in version 11.01 and you updated only your "working version"? I think this paragraph should be rephrased to discuss in detail which updates of the CO-only simulation has been performed by whom (in which version).

The revision between manuscript versions better reflects relationship to the standard model.

When we began this work (using v9-01-03), the updates described in that paragraph did not yet exist and so we implemented them in our working version – but also sent them to the GEOS-Chem support team. As objective improvements, they were added to the standard version of the model before our manuscript was accepted. These are small updates to the code that have already been included in the current (and all future) standard versions of the simulation, so we feel further discussion here is unnecessary and would be distracting.

Technical correction:
Supplement: Fig S8 - Figure and caption are on different pages.

We have moved the caption to the same page as the figure (Fig. S9).

**Manuscript changes**

1. Added reference to new Fig. S3 (P6 L26 and P17 L5)
2. Renumbered subsequent figures in the supplement and updated references in main text
3. Expanded Supplement Section S1 (see above)
4. Added Supplement Figure S3.

[revised manuscript text omitted]